# A Health Care Professional Delivered Low Carbohydrate Diet Program Reduces Body Weight, Haemoglobin A1c, Diabetes Medication Use and Cardiovascular Risk Markers—A Single-Arm Intervention Analysis

**DOI:** 10.3390/nu14204406

**Published:** 2022-10-20

**Authors:** Grant D. Brinkworth, Thomas P. Wycherley, Pennie J. Taylor, Campbell H. Thompson

**Affiliations:** 1Commonwealth Scientific and Industrial Research Organisation (CSIRO)—Health and Biosecurity, Westmead, NSW 2145, Australia; 2Alliance for Research in Exercise, Nutrition and Activity, University of South Australia, Adelaide, SA 5000, Australia; 3Department of Medicine, University of Adelaide, Adelaide, SA 5000, Australia

**Keywords:** diet, lifestyle, weight loss, glycaemic control, diabetes

## Abstract

This study examined the effectiveness of a health care professional delivered low-carbohydrate diet program (Diversa Health Program) aiming to improve obesity/type-2-diabetes management for people living in Australia. 511 adults (Age:57.1 ± 13.7 [SD] yrs) who participated between January 2017–August 2021 for ≥30 days with pre-post data collected for ≥1 key outcome variable (body weight and HbA1c) were included in the analysis. Average participation duration was 218 ± 207 days with 5.4 ± 3.9 reported consultation visits. Body weight reduced from 92.3 ± 23.0 to 86.3 ± 21.1 kg (*n* = 506, *p* < 0.001). Weight loss was 0.9 ± 2.8 kg (1.3%), 4.5 ± 4.3 kg (5.7%) and 7.9 ± 7.2 kg (7.5%), respectively, for those with a classification of normal weight (*n* = 67), overweight (*n* = 122) and obese (*n* = 307) at commencement. HbA1c reduced from 6.0 ± 1.2 to 5.6 ± 0.7% (*n* = 212, *p* < 0.001). For members with a commencing HbA1c of <5.7% (*n* = 110), 5.7–6.4% (*n* = 55), and ≥6.5% (*n* = 48), HbA1c reduced −0.1 ± 0.2%, −0.3 ± 0.3%, and −1.4 ± 1.3%, respectively. For members with a commencing HbA1c ≥6.5%, 90% experienced a HbA1c reduction and 54% achieved a final HbA1c < 6.5%. With inclusion and exclusion of metformin, respectively, 124 and 82 diabetes medications were prescribed to 63 and 42 members that reduced to 82 and 35 medications prescribed to 51 and 26 members at final visit. A health care professional delivered low-carbohydrate diet program can facilitate weight loss and improve glycaemic control with greatest improvements and clinical relevance in individuals with worse baseline parameters.

## 1. Introduction

Worldwide and in Australia, 460 million and 1.2 million people, respectively, are estimated to have type 2 diabetes (T2D) [1]. The World Health Organisation estimates 39% of adults worldwide are overweight and 13% are obese; in Australia, 67% of adults (~12.5 million) are estimated to be either overweight or obese [2]. With rates of T2D, overweight and obesity reaching epidemic levels that continue to grow, public health and economic systems are presented with a major challenge [3,4].

Diet remains the first-line treatment of obesity and T2D. Traditionally, public practice guidelines promoted an energy reduced, high carbohydrate, low fat (HC) diet for obesity and T2D management [5,6,7]. However, several recently published systematic reviews and meta-analyses of well-controlled studies conducted in adults with T2D have shown that compared to a traditional HC diet, a low carbohydrate, high fat (LC) diet promotes greater weight loss, improvement in blood glucose control and reduction in diabetes medication requirements [8,9,10,11,12,13]. However, availability of these lifestyle interventions have been largely limited to patients participating in the research studies and few publicly available LC diet programs, targeting obesity and T2D management, are available and have been evaluated.

Health care professionals (HCPs) remain the centre point of effective management of T2D [14]. Previous studies have reported weight loss and improved glycaemic control in individuals with T2D with the use of self-management digital and online delivered LC diet programs [15,16]. However, evaluation of public accessible HCP delivered LC programs that integrate health coaches and medical support are required.

The Diversa Health Care Program is a paid subscription-based health service that delivers a personalised LC plan by HCPs including general practitioners, health educators and allied health professionals who provide education, medical monitoring, and supervised care via telehealth. Additionally, members are provided real-time access and virtual group support. The programs aims to improve weight and T2D management for people living in Australia. The purpose of this study was to assess the effectiveness of the Diversa Health Care Program for the management of body weight and metabolic and diabetes risk factors. Specifically, the primary endpoints to assess effectiveness of the program were changes in body weight and glycosylated haemoglobin (HbA1c) at the participants final recorded program consultation visit. Secondary endpoints assessed included blood pressure, waist circumference and blood lipid levels.

## 2. Methods

### 2.1. Study Setting, Participants, and Design

This single-arm, pre-post intervention study design, retrospectively analysed data from individuals aged ≥18 years who joined the Diversa Health Program (a private-for-profit entity previously known as ‘The Low Carb Clinic’; VIC, Australia); and paid for their services covering a period of January 2017–August 2021. Individuals (termed members) were self-referred via word of mouth or advertising or referred by physicians to join the program primarily for the purposes of body weight management, diabetes prevention or management, or to address other metabolic health conditions.

Member data were included in the analysis if the member remained in the program for a minimum duration of 30 days (one month) and had at least 2 consultation visits with a health care professional of the Diversa Health Care team, irrespective of data availability for all reported outcomes. For example, member data for weight were included in the absence of availability of HbA1c data, and vice versa. Baseline data were classified as available data recorded at or within 30 days of the first program consultation. Post-program data were classified as the data available for the member at their most recent consultation, or when they were deemed to have discontinued the program. If a member had a gap between program consultation visits of >395 days (1 year plus 30 days) the member was deemed to have discontinued the program and the longer period of their program data was used for analysis (i.e., the period from initial program consultation to the last consultation before the >395 day gap, or the period from when the participant re-commenced the program (if applicable) to their most recent consultation visit). In the latter scenario, the program recommencement consultation measures were classified as the member’s baseline values.

### 2.2. Intervention

The Diversa Health Care Program is a HCP delivered program providing remote consultations, unlimited technology-enabled communication support, and a personalised LC diet plan, targeting individuals with or at risk of diabetes living in Australia. The user initially subscribes to a membership plan at a minimum monthly cost of $269, $229 or $209 AUD with a 3-, 6- and 12-month commitment, respectively. At initiation, members complete a consultation with a health coach (either an accredited practicing dietitian, registered nurse, or experienced health educator) to set goals, discuss dietary requirements, support needs and implications of a LC diet lifestyle; and medical practitioner to undergo a medical history assessment, review clinical and biochemical blood test results and medication history and implications with participating in the program. The LC diet eating plan is delivered using a personalised approach that emphasises eating to satiety with ad libitum intake of dietary protein and fat and limiting dietary carbohydrate intake to <30 g/day to achieve a macronutrient intake profile of approximately 5% carbohydrate, 35% protein and 60% fat (sample daily menu plan presented in Table 1). Throughout the program, participants have access to a variety of voluntary support functions and program features but are required to purchase and prepare their own food. These include:**Medical Monitoring**—Medical review with a Diversa Health General Medical Practitioner at a minimum frequency of every 3 months to review pathology results and make necessary medication adjustments.**One-on-One Coaching Real-Time Support**—Live progress review and consultation with a health coach at least monthly and up to weekly. Ongoing support and advice from a health coach via SMS, WhatsApp, Facebook and Email, as required.**Online member Portal**—Access to an online portal in which participants can access/track pathology results, share notes with a health coach and chat via a Live Chat function, and access education resources personally selected by the health coach.**Community**—Access to a range of health coach-led group webinars, with option to learn, connect and share resources with other members via a moderated Facebook community. Family members are provided access to a family-based webinar to support the members journey.**Resources**—Food guides including information of what to eat, meal planning ideas and shopping advice to tailor the dietary plan to individual dietary requirements and sensitivities. Recipes and information fact sheets personally selected by the members health coach.

### 2.3. Data Collection and Study Outcomes

The Diversa Health program team were responsible for the collection and storage of member data, including registration details and clinical information. At each consultation visit, medication prescription was recorded. Additional information including height, body weight, waist circumference, blood pressure, and biochemical data (blood glucose, insulin, HbA1c, blood lipids and liver function markers [alanine aminotransferase—ALT, aspartate aminotransferase—AST, γ-glutamyltransferase—GTT) were attempted to be collected from the participants who voluntarily provided self-reported information or pathology results initiated by the Diversa Health care team or obtained from health checks conducted by their overseeing general practitioner ‘outside’ of the Diversa Health Care team. Established criteria of overweight and obesity by body mass index (BMI) defined by the World Health Organisation [17] [Underweight: <18.5 kg/m^2^; Normal Weight: 18.5–24.9 kg/m^2^; Overweight: 25–29.9 kg/m^2^; Class 1 Obesity: 30–34.9 kg/m^2^; Class 2 Obesity: 35–39.8 kg/m^2^; Class 3 Obesity: 40+ kg/m^2^], and diabetes status by HbA1c level defined by the American Diabetes Association [18] [Normoglycemia: <5.7%; Pre-diabetes: 5.7–6.4%; Diabetes ≥6.5%], were used in the analysis for member categorisation. Achievement of diabetes remission was determined using established criteria (HbA1c of <6.5% for at least three months after stopping glucose-lowering medication) established by an expert consensus report [19].

Deidentified data were provided to the research team. As part of the program joining process, individuals participating in the Diversa Health program provided consent to use their data for research purposes and therefore no consent was obtained directly from each individual to undertake this specific analysis and study. The study was approved by the Commonwealth Scientific and Industrial Research Organization (CSIRO) Human Research Ethics Committee (2022_011_LR).

### 2.4. Statistical Analysis

All statistical analyses were conducted using SPSS statistical software (Version 28.0, Armonk, NY, USA). Descriptive statistics calculated included means and standard deviations of continuous variables and proportions for categorical variables. Paired *t*-tests were used to compare changes in weight, waist circumference, blood pressure, HbA1c, blood glucose, insulin, lipids, and liver function tests. Statistical significance was set at *p* < 0.05 (two-tail). Of the 511 members that met the data inclusion criteria, 279 (54.6%) had reported data for at least one of the biochemical outcomes assessed; HbA1c, blood glucose, insulin, lipids, and liver function. Consequently, 45.4% (*n* = 232) of the members included in the analysis had biochemical data missing that could not be analysed. Given the potential for those with and without biochemical data reported to represent different population groups, data for changes in body weight, waist circumference and blood pressure changes were analysed and separately reported for both the entire member group, as well as for the sub-group who reported biochemical data.

## 3. Results

### 3.1. Characteristics

Data were made available for 697 individuals who completed the program sign up process. 511 members met the inclusion criteria and had a program duration of at least 30 days with at least 2 consultation visits. Majority of the members were female (296 female (58%), 145 males (28%), 70 (14%) not reported), with an average age of 57.1 ± 13.7 years (n = 455 reported) and participated in the program for an average duration of 218 ± 207 days and had attended an average 5.4 ± 3.9 HCP consultations.

### 3.2. Body Weight

503 members had data for body weight at initial consultation and during the program (Table 2). Initial average body weight was 92.3 ± 23.0 kg and average weight loss was 6.0 ± 6.6 kg (6.5% [males: 4.9%; females 6.8%]) over an average duration of 216 days (Table 2, Figure 1). 69.4% (*n* = 349), 49.3% (*n* = 248) and 22.5% (*n* = 113) achieved a weight loss of 3%, 5% and 10%, respectively. From initial consultation to final visit, changes in normal, overweight and obesity classifications were: normal weight (14% to 22%), overweight (25% to 29%), class 1 obesity (27% to 26%), class 2 obesity (18% to 13%) and class 3 obesity (17% to 10%). For members with a starting BMI ≥ 25 kg/m^2^ (classified as overweight or obese; *n* = 429), overall weight loss was −6.9 ± 6.7 kg (−7.0 ± 6.4%); 75.5% (*n* = 324) achieved 3% weight loss, 54.3% (*n* = 233) achieved 5% weight loss and 25.4% (*n* = 109) achieved 10% weight loss.

For all members who reported data for at least one of the biochemical outcomes (HbA1c, blood glucose, insulin, lipids, and liver function) and body weight at the initial consultation and during the program (*n* = 277, Table 2), members had an initial average body weight of 90.6 ± 22.1 kg and achieved an average weight loss of 7.2 ± 7.3 kg (8.0%) over an average duration of 256 days. 72.9% (*n* = 202), 58.8% (*n* = 163) and 30.0% (*n* = 83) of members achieved a weight loss of 3%, 5% and 10%, respectively. From initial consultation to final visit, changes in body weight classifications were: normal weight (14% to 26%), overweight (27% to 32%), class 1 obesity (31% to 26%), class 2 obesity (15% to 11%) and class 3 obesity (14% to 6%). For these members with a starting BMI ≥ 25 kg/m^2^, overall weight loss was 8.2 ± 7.3 kg (8.4 ± 7.1%). 79.3% (*n* = 188), 66.2% (*n* = 157) and 34.6% (*n* = 82) achieved a weight loss of 3%, 5% and 10%, respectively.

Members classified as having normoglycemia, pre-diabetes or diabetes by HbA1c level at initial consultation experienced a weight loss of 7.9%, 7.4% and 8.5%, respectively, over an average duration range of 252, 262 to 269 days, respectively.

### 3.3. HbA1c and Diabetes Status

At program commencement, members who reported HbA1c at the initial consultation and during the program (*n* = 212) had an average HbA1c of 6.0 ± 1.2%. Based on HbA1c classification levels, 52% (*n* = 110) had normoglycemia, 26% (*n* = 54) had pre-diabetes, and 23% (*n* = 48) had diabetes. At final visit, average reductions in HbA1c were −0.4% (all members combined [males: −0.5%, females: −0.4%]), −0.1% (normoglycemia), −0.3% (pre-diabetes) and −1.4% (diabetes), Table 3, Figure 2. For members commencing with a HbA1c between ≥5.7–6.4% (pre-diabetes classification), at final visit 78% (42/54), 15% (8/54) and 7% (4/54) had experienced a decrease, increase or no change in HbA1c levels, respectively; and 48% (26/54) had reduced HbA1c level to below 5.7%. For members commencing with a HbA1c ≥ 6.5% (diabetes classification), at final visit 90% (43/48), 9% (4/48), and 1% (1/48) had experienced a decrease, increase or no change in HbA1c levels, respectively; 54% (26/48) had reduced HbA1c level to below 6.5%; and 27% (13/48) achieved diabetes remission. At final visit based on HbA1c classification levels, 66.0% (*n* = 140) had normoglycemia, 23.1% (*n* = 49) had pre-diabetes and 10.8% (*n* = 23) had diabetes.

### 3.4. Metabolic Risk Factors

At final visit, there were statistically significant reductions in population average waist circumference, blood pressure, blood glucose, insulin, triglycerides and liver function markers, and significant increases in average HDL-cholesterol, total and LDL cholesterol (*p* < 0.05 for all, Table 2 and Table 3).

### 3.5. Medication

At baseline, a total number of 63 members were prescribed 124 dosages of diabetes medications (including metformin), and 42 members prescribed 82 medication dosages (excluding metformin) (Table 4). Throughout the program, dosage changes of diabetes medications across the subclasses for prescribed users were reported, including medication eliminations, decreases, increases, commencements, or no changes. The proportion of these changes classified as a either a diabetes medication dosage elimination or decrease was 47.5% (including metformin) and 62.8% (excluding metformin) (Figure 3). At final visit the total number of members prescribed diabetes medication and total dosages prescribed had reduced to 51 members being prescribed 82 dosages (including metformin) and 26 members being prescribed 35 medication dosages (excluding metformin) (Table 4).

## 4. Discussion

This study demonstrates a HCP delivered LC diet program is feasible and effective for achieving clinically relevant weight loss, improvements in blood glucose control and reductions in cardiovascular disease risk factors for individuals who are overweight or obese and/or with elevated blood glucose levels. The changes observed have been associated with reduced risk of obesity comorbidities, diabetes-related complications, cardiovascular disease, and health care costs [20,21,22,23,24,25]. Marked discontinuation or reductions in the usage of diabetes medications were also observed.

### 4.1. Weight Loss

An overall, 6.0% mean weight loss in the entire study sample was observed after an average program participation duration of ~7 months. Similarly, an average 8.0% weight loss over an 8.5 month period was achieved in the member sub-group who also provided biochemical data. Sensitivity analysis revealed a −7.0% and −8.4% body weight reduction in the entire study sample and sub-sample, respectively, following exclusion of members who had a normal body weight classification at baseline. This is comparable with other commercial weight loss programs delivered by one-on-one meetings with trained consultants including telehealth and e-mail contacts [26], and greater than that previously reported with commercial online programs [27,28,29].

A 5% weight loss is considered clinically meaningful to reduce the risk of obesity-related comorbidities and health care costs [20]. Although there is a continuum of clinically meaningful weight loss, which varies among individuals and depends on specific comorbidities and complications. Sustained weight reductions of as little as 2–5% have shown significant benefits in cardiovascular risk factors [30], and greater weight loss has been associated with greater improvements in obesity-related comorbidities with a reduction in mortality with weight loss >10% reported [20]. Across the entire member cohort, 50% achieved a weight loss of at least 5% and 23% achieved a weight loss of at least 10%. Correspondingly, there was a systematic downward shift across overweight and obese weight classifications, such that the number of participants across the entire sample classified as obese reduced from 62% at baseline to 49% at final visit, with number of participants classified as normal weight increasing from 14% to 22%. It is well accepted a high BMI increases mortality risk, with the BMI range of 18.5 to <25 kg/m^2^ generally associated with lowest mortality risk, and risk increasing as BMI increases throughout the range of BMIs classified as overweight and obese [31,32].

### 4.2. Blood Pressure

In response to the weight loss achieved, the substantial reductions in blood pressure (−9/−4 mmHg) observed have been associated with clinically significant reduction in risks of diabetes-relations complications, cardiovascular disease, and mortality [20]. Reboldi et al. [33] reported a 13% reduction in stroke risk for each 5 mmHg reduction in systolic blood pressure and an 11.5% reduction in stroke risk for each 2 mmHg reduction in diastolic blood pressure.

### 4.3. HbA1c

Across the study sample, an absolute HbA1c reduction of 0.4% occurred after 8 months. More specifically, members with a baseline HbA1c level above the diabetes classification [34], experienced a 1.4% HbA1c reduction that is substantially greater than a previous small study that reported an 0.8% absolute reduction in HbA1c in adults with T2D after 32 weeks participation of an online delivered LC diet intervention [16]. Other studies that have administered LC diet programs to patients with T2D using an in-person model for at least 3 months have reported HbA1c reductions between 0.6−1.6% [35,36,37,38,39,40,41,42]. A 0.3% HbA1c reduction is considered a clinically meaningful to reduce diabetes related long-term complications [22,23]. Moreover, the large 1.4% HbA1c reduction observed in those with a baseline HbA1c above the diabetes classification exceeds the standards set by the FDA for approval of pharmacotherapy diabetes treatment [43].

At final visit, the prevalence members with a HbA1c classified as normoglycemic increased and as diabetes declined. Specifically, ~50% of members initially classified as having pre-diabetes and diabetes HbA1c levels at program commencement had lowered their HbA1c levels to a point where they were no longer within the pre-diabetes and diabetes classification, respectively. This result is comparable with the proportion of individuals with T2D who achieved diabetes reversal after 1 year participation in an intensive, digitally monitored continuous care program, including individualised support with telemedicine, health coaching, and guidance using an individualised LC diet [44]. The Diabetes Prevention Program (DDP) Outcomes study showed that reversion to normoglycemia, even if transient, is associated with reduced risk of diabetes [45]. Moreover, the UK Prospective Diabetes Study demonstrated a legacy effect of a period of good blood glucose control over a 10 year follow up in patients with T2D [46]. This suggests that even if diabetes and poor blood glucose control recurs, reversal of pre-diabetes and diabetes status and improvements in blood glucose control observed in individuals participating in the program can translate to reduced risk of future diabetes and/or vascular events.

A large systematic review showed weight loss in patients with T2D is consistently accompanied by HbA1c reduction in a dose-dependent manner (0.1% HbA1c reduction for every 1 kg weight loss) for overall populations [47]; however HbA1c lowering was greater in populations with poor glycaemic control than in well controlled populations with the same degree of weight loss. Similarly, the present results showed stepwise greater HbA1c reductions in individuals classified as normoglycemic, pre-diabetes and diabetes despite comparable weight loss. Despite this, members with a baseline HbA1c above the diabetes classification cut-off who still had relatively good average glycaemic control (7.9%), achieved a HbA1c reduction that far exceeded the level that could have been expected for the corresponding level of weight loss.

A separate body of evidence has demonstrated in individuals with T2D that reducing dietary carbohydrate can reduce HbA1c levels independent of weight loss [48]. A systematic review and meta-analysis of 10 RCTs conducted in adults with T2D reported greater HbA1c reductions following a LC diet compared to traditional high carbohydrate diet, and that a greater glucose-lowering effect was related to a lower carbohydrate intake [8]. This suggest the prescription of a LC diet in the Diversa Health program may have contributed to the blood glucose lowering effects observed. Collectively, these findings suggest this program and the use of a LC diet may have the greatest clinical relevance and benefits in those with elevated blood glucose levels, prediabetes and T2D.

### 4.4. Medication Changes

In addition to the decreases in HbA1c levels, concurrent reductions in medications prescribed for blood glucose-lowering medications were reported. Across all diabetes medication classes (including metformin), 63% of the medication dosage changes were either reductions or cessations across the observed participation period. Notably, of the insulin dosage changes reported, 68% were cessations and 18% were reductions. Due to the treatment costs and risk of side effects, including symptomatic hypoglycaemia and weight gain, often observed with drug related glycaemic control [49,50], the treatment approach utilised in this program may be more advantageous to achieve blood glucose management for improved health outcomes with reduced risk compared to drug treatment. Conduct of a health economic analysis would assist to understand the cost effectiveness of the program examined.

### 4.5. Blood Lipids

The increase in HDL cholesterol and reduction in triglycerides observed is consistent with previously published systematic reviews of randomised controlled trials examining the effects of LC diets [9,10,11,12,13]. These results suggest that a LC diet is an effective approach to improve lipid abnormalities associated with insulin resistance and the metabolic syndrome, which increases CVD risk in T2D [51]. Previous studies have reported that every 1% increase in HDL cholesterol and 1 mmol/L reduction in triglycerides, may constitute a 3% or 8–21% reduction in heart disease risk, respectively [24,25]. Therefore, the 8% increase in HDL cholesterol and 0.23 mmol/l reduction in triglycerides observed in the participating in this program could translate into a substantial reduction in CVD risk.

Despite the favourable changes in HDL cholesterol and triglycerides levels, average LDL cholesterol levels increased at the final visit of the program. This is similar to some [16,39,52,53,54], but not all [40,55], previous studies that have reported increases in LDL cholesterol with a LC diet. The health effects of any increase in LDL cholesterol in response to consumption of a LC diet has been extensively discussed previously. It is considered that LDL cholesterol particle size may be a stronger predictor of heart disease risk rather than LDL per se [56]. Separate studies have shown that a LC diet tends to increase the size of LDL cholesterol particles and reduce the number of smaller LDL cholesterol particles [57,58], which is considered to reduce heart disease risk [56,59,60]. This suggests the possibility that the increase in LDL cholesterol observed may promote a cardioprotective effect. Further longer-term studies are required to examine the long-term effects on cardiovascular events.

### 4.6. Liver Function Markers

The observed reduction in the level of liver function enzymes post program is consistent with several previous studies that have shown that weight loss improves blood markers of liver disease including ALT and AST [61,62,63]. Previous studies have reported that elevated levels of liver enzymes are positively associated with cardiovascular risk factors, metabolic syndrome and T2D risk [64,65,66].

### 4.7. Strengths and Limitations

A strength of the study was the broad inclusion of individuals across varying body weights and blood glucose status categories with limited exclusion criteria for program participation, which promotes broad generalisability of the study results. However, it should be acknowledged the program operates on a user-pay delivery model requiring out of pocket payments by members for consultations with HCPs and health services. This may limit representation of individuals with insufficient resources to pay for program access and lower socio-economic populations, who have a higher risk of pre-diabetes and T2D [67]. Further research is required to understand how this program can be embedded into routine health services to achieve wide-scale adoption. The real-world nature of the intervention is a study strength; however, sporadic data reporting creates difficulties in making direct comparisons between individuals and outcomes measures. This highlights the importance of careful consideration in establishing systemic data collection approaches within public health interventions. The single-arm design with no control group and the lack of dietary intake data is a limitation of the study that precludes the ability to determine the specific components of the program/intervention responsible for the changes observed. The study also lacks follow-up data and adverse events were not systematically recorded making it difficult to determine any side-effects from participating in the health program evaluated. Despite often-cited concerns of side-effects with the use a LC diet [68], a recent systematic review reported no significant or clinically important increase in total or serious adverse events in individuals following a LC diet [13]. The relatively short-duration of the program evaluation also limits the understanding of the long-term sustainability and health effects of the changes in weight, blood glucose control and cardiovascular disease risk makers observed. Further long-term studies are required using a randomised controlled trial design that examine clinical endpoints such as diabetes and cardiovascular events that will provide greater understanding of the therapeutic potential of the program evaluated.

## 5. Conclusions

The present results indicate that an HCP delivered LC diet program can facilitate clinically relevant weight loss and improved blood glucose control in individuals who are overweight or obese and/or with elevated blood glucose levels. This resulted in an overall reversion across obesity and glycaemic levels for diabetes classifications and reductions in diabetes medication use. Further studies are required to determine the longer-term sustainability and health effects of this program and it could be integrated mainstream medical care to have wider impact in the treatment of obesity and T2D.

## Figures and Tables

**Figure 1 nutrients-14-04406-f001:**
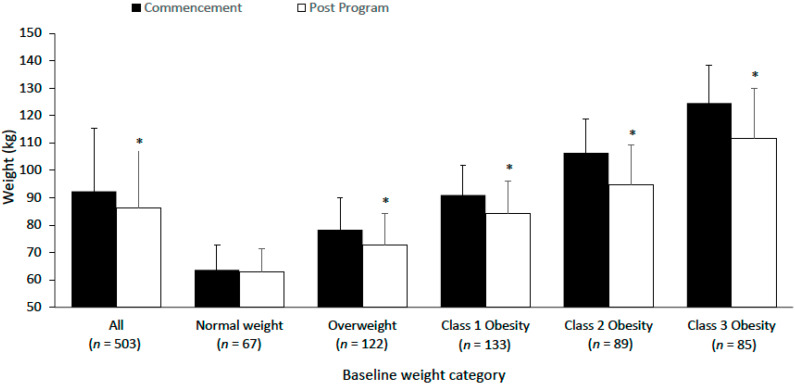
Body weight before and at final visit for all members and by baseline weight categorisation of normal weight (BMI: 18.5–24.9 kg/m^2^), overweight (25–29.9 kg/m^2^), class 1 obesity (30–34.9 kg/m^2^), class 2 obesity (35–39.9 kg/m^2^), class 3 obesity (40+ kg/m^2^). Values are mean ± standard deviation. * *p* < 0.001 significantly lower compared to commencement.

**Figure 2 nutrients-14-04406-f002:**
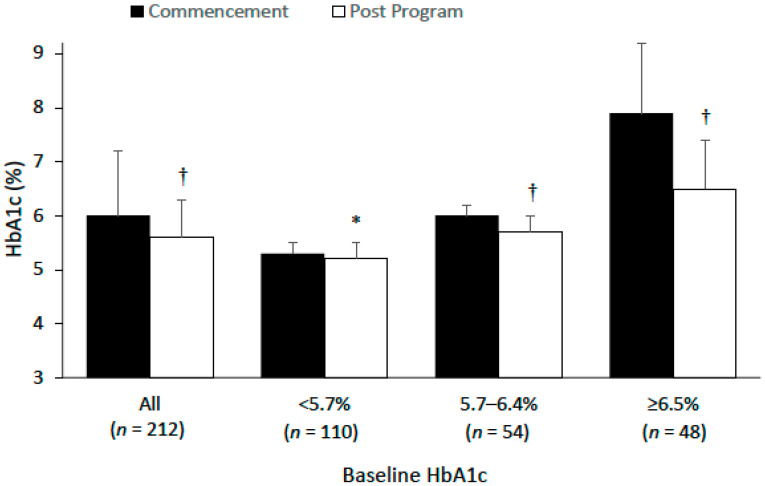
Haemoglobin A1c (HbA1c) before and at final visit for all members and for those with a baseline HbA1c of <5.7%, 5.7–6.4% and ≥6.5%. Values are mean ± standard deviation. * *p* < 0.05, † *p* < 0.001 significantly lower compared to commencement.

**Figure 3 nutrients-14-04406-f003:**
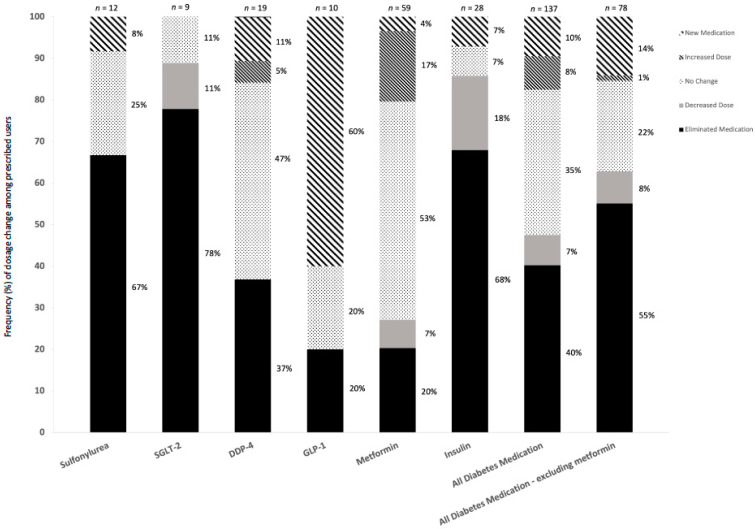
Frequency of medication dose changes by drug class. Bars represent total number and relative frequency of reported dose changes.

**Table 1 nutrients-14-04406-t001:** Two-day sample meal plan for the Diversa Health Low Carbohydrate Diet eating plan.

Mealtime	Day 1 Total Daily Intake Excluding Snacks: 24 g Carbohydrate, 161 g Fat, 150 g Protein	Day 2 Total Daily Intake Excluding Snacks: 17 g Carbohydrate, 232 g Fat, 84 g Protein
**Breakfast**	**Bacon, Spinach and Mushroom omelette**: - 3 eggs - 50 g diced bacon - 50 g sliced mushrooms - 2 tsp butter - 50 g tasty cheese - ½ cup spinach - Pinch of salt	**Coconut and Chia Pudding**: - 100 g coconut cream - 6 blueberries - ¼ cup chia seeds - 20 g toasted coconut
**Lunch**	**Bacon, Spinach and Mushroom omelette**: - 3 eggs - 50 g diced bacon - 50 g sliced mushrooms - 2 tsp butter - 50 g tasty cheese - ½ cup spinach - Pinch of salt **Plus side of lemon infused green beans** - Drizzle olive oil - ¼ cup fresh lemon juice - 1 tbsp seeded mustard - 200 g green beans - Salt and pepper to taste	**Pork with Blue Cheese Sauce** - 75 g blue cheese - 100 mL cream - 2 pork chops - 100 g fresh green beans - 1 tbsp butter - Salt and pepper to taste
**Dinner**	**Creaming baked Atlantic salmon on buttery cauliflower mash:** - 1 fillet of salmon with skin (150–200 g) - 2 tsp minced garlic - 2 tsp dried basil - 1 tbsp butter - 1 wedge of fresh lemon - 250 g fresh cauliflower - 15 mL olive oil	**Bacon, Lettuce, Tomato (BLT) Creamy Salad** - 150g chicken breast - ½ rasher middle bacon - 1 egg - ½ large tomato - ¼ onion - 1 cup mixed salad - 15 g mayonnaise, no added sugar - 2 tsp lemon juice - Pepper to taste
**Snacks (optional)**	2 hardboiled eggs	30 g prosciutto

g—grams; tbsp—tablespoon; tsp—teaspoon.

**Table 2 nutrients-14-04406-t002:** Body weight, waist circumference and blood pressure data at commencement (initial consultation) and post program (final visit) of the Diversa Health program for all members.

Variable	Initial Consultation—All Participants		Initial Consultation—Participants with Pre/Post Program Data		Final Visit	Change	*p* Value	Duration from Initial Consultation to Last Recorded Observation—Days
	*N* (%)	Mean (SD)	*N* (%)	Mean (SD)	Mean (SD)	Mean (SD)		Mean (SD)
Initial Consultation	511							218.1 (207.5)
Age (years)	455	57.1 (13.7)	-	-	-	-	-	-
Gender	-	-	-	-	-	-	-	-
Female	296 (57.9)	-	-	-	-	-	-	-
Male	145 (28.4)	-	-	-	-	-	-	-
Not reported	70 (13.7)	-	-	-	-	-	-	-
Weight (kg)	506	92.2 (23.0)	503	92.3 (23.0)	86.3 (21.1)	−6.0 (6.6)	<0.001	216.2 (203.2)
BMI (kg/m^2^)	499	32.8 (7.6)	496	32.9 (7.6)	30.7 (6.9)	−2.2 (2.4)	<0.001	217.6 (203.8)
Weight (kg) by weight category	499		496					
Underweight BMI: <18.5 kg/m^2^	0 (0)	N/A	0 (0)	N/A	N/A	N/A	N/A	N/A
Normal Weight BMI: 18.5–24.9 kg/m^2^	69 (13.8)	63.5 (8.4)	67 (13.5)	63.5 (8.4)	62.6 (8.2)	−0.9 (2.8)	0.443	252.9 (260.9)
Overweight Weight BMI: 25–29.9 kg/m^2^	122 (24.4)	77.9 (10.7)	122 (24.6)	77.9 (10.7)	73.4 (10.5)	−4.5 (4.3)	0.010	208.6 (185.7)
Class 1 Obesity BMI: 30–34.9 kg/m^2^	134 (26.9)	91.0 (11.3)	133 (26.8)	91.0 (11.3)	85.0 (12.2)	−6.0 (5.5)	<0.001	233.7 (211.4)
Class 2 Obesity BMI: 35–39.9 kg/m^2^	89 (17.8)	105.0 (11.6)	89 (17.9)	105.0 (11.6)	96.3 (12.7)	−8.7 (7.0)	<0.001	205.9 (191.5)
Class 3 Obesity BMI: 40+ kg/m^2^	85 (17.0)	125.6 (17.9)	85 (17.1)	125.6 (17.9)	115.6 (19.2)	−10.0 (8.9)	<0.001	189.6 (175.0)
Weight status classification	499		496					
Underweight BMI: <18.5 kg/m^2^	0 (0)	-	0 (0)	-	1 (0.2)	-	-	-
Normal Weight BMI: 18.5–24.9 kg/m^2^	69 (13.8)	-	67 (13.5)	-	108 (21.8)	-	-	-
Overweight Weight BMI: 25–29.9 kg/m^2^	122 (24.4	-	122 (24.6)	-	143 (28.8)	-	-	-
Class 1 Obesity BMI: 30–34.9 kg/m^2^	134 (26.9)	-	133 (26.8)	-	129 (26.0)	-	-	-
Class 2 Obesity BMI: 35–39.9 kg/m^2^	89 (17.8)	-	89 (17.9)	-	64 (12.9)	-	-	-
Class 3 Obesity BMI: 40+ kg/m^2^	85 (17.0)	-	85 (17.1)	-	51 (10.3)	-	-	-
Waist Circumference	424	104.2 (17.4)	397	104.8 (17.2)	99.1 (15.9)	−5.7 (6.9)	<0.001	210.6 (194.9)
Systolic Blood Pressure (mmHg)	304	138.2 (20.6)	256	139.6 (20.6)	131.0 (18.7)	−8.6 (18.2)	<0.001	225.4 (211.9)
Diastolic Blood Pressure (mmHg)	304	84.6 (12.7)	256	84.9 (13.1)	81.2 ± 13.1	−3.7 ± 12.1	<0.001	225.4 ± 211.9

BMI—Body Mass Index.

**Table 3 nutrients-14-04406-t003:** Body weight, waist circumference, blood pressure and metabolic risk markers at commencement (initial consultation) and post program (final visit) of the Diversa Health program for members with reported biochemical data.

Variable	Initial Consultation—All Participants		Initial Consultation—Participants with Pre/Post Program Data		Final Visit	Change	*p* Value	Duration from Initial Consultation to Last Recorded Observation—Days
	*N* (%)	Mean (SD)	*N* (%)	Mean (SD)	Mean (SD)	Mean (SD)		Mean (SD)
Initial Consultation	279							258.7 (206.8)
Age (years)	274	59.4 (13.0)	-	-	-	-	-	-
Gender			-	-	-	-	-	-
Female	176 (63.1)							
Male	91 (32.6)							
Not reported	12 (4.3)							
Weight (kg)	277	90.6 (22.1)	277	90.6 (22.1)	83.4 (19.8)	−7.2 (7.3)	<0.001	256.0 (201.6)
BMI (kg/m^2^)	274	32.1 (7.1)	274	32.1 (7.1)	29.5 (6.3)	−2.6 (2.7)	<0.001	258.4 (201.5)
Weight (kg) by weight category	274		274					
Underweight BMI: <18.5 kg/m^2^	0 (0)	N/A	0 (0)	N/A	N/A	N/A	N/A	N/A
Normal Weight BMI: 18.5–24.9 kg/m^2^	37 (13.5)	63.7 (9.1)	37 (13.5)	63.7 (9.1)	62.9 (8.5)	−0.8 (2.6)	0.071	263.4 (242.9)
Overweight Weight BMI: 25–29.9 kg/m^2^	74 (27.0)	78.3 (11.6)	74 (27.0)	78.3 (11.6)	72.7 (11.5)	−5.6 (4.6)	<0.001	245.4 (162.1)
Class 1 Obesity BMI: 30–34.9 kg/m^2^	84 (30.7)	91.0 (11.0)	84 (30.7)	91.0 (11.0)	84.2 (12.0)	−6.8 (6.0)	<0.001	280.2 (228.6)
Class 2 Obesity BMI: 35–39.9 kg/m^2^	41 (15.0)	106.4 (12.3)	41 (15.0)	106.4 (12.3)	94.9 (14.3)	−11.5 (8.0)	<0.001	225.5 (168.9)
Class 3 Obesity BMI: 40+ kg/m^2^	38 (13.9)	124.6 (18.3)	38 (13.9)	124.6 (18.3)	111.7 (19.5)	−12.9 (9.9)	<0.001	245.4 (162.1)
Weight (kg) by glycaemic control classification	259		259					
Normoglycemia HbA1c: <5.7%	158 (61.0)	91.7 (22.7)	158 (61.0)	91.7 (22.7)	84.6 (20.2)	−7.2 (7.3)	<0.001	251.7 (206.5)
Pre-diabetes HbA1c: 5.7–6.4 %	55 (21.2)	87.9 (21.1)	55 (21.2)	87.9 (21.1)	81.4 (19.4)	−6.5 (6.0)	<0.001	261.8 (171.3)
Diabetes HbA1c: ≥6.5%	46 (17.8)	89.2 ± 22.0	46 (17.8)	89.2 (22.0)	81.6 (19.4)	−7.6 (7.6)	<0.001	269.1 (178.4)
Weight status classification	274		274					
Underweight BMI: <18.5 kg/m^2^	0 (0)	-	0 (0)	-	1 (0.4)	-	-	-
Normal Weight BMI: 18.5–24.9 kg/m^2^	37 (13.5)	-	37 (13.5)	-	70 (25.5)	-	-	-
Overweight BMI: 25–29.9 kg/m^2^	74 (27.0)	-	74 (27.0)	-	87 (31.8)	-	-	-
Class 1 Obesity BMI: 30–34.9 kg/m^2^	84 (30.7)	-	84 (30.7)	-	70 (25.5)	-	-	-
Class 2 Obesity BMI: 35–39.9 kg/m^2^	41 (15.0)	-	41 (15.0)	-	29 (10.6)	-	-	-
Class 3 Obesity BMI: 40+ kg/m^2^	38 (13.9)	-	38 (13.9)	-	17 (6.2)	-	-	-
Waist Circumference	228	103.9 (16.4)	220	104.5 (16.1)	97.7 (14.6)	−6.8 (7.2)	<0.001	246.3 (191.9)
Systolic Blood Pressure (mmHg)	186	140.2 (21.5)	167	141.0 (21.1)	131.6 (19.3)	−9.3 (19.6)	<0.001	234.0 (185.9)
Diastolic Blood Pressure (mmHg)	186	84.7 (13.6)	167	85.0 (13.8)	81.0 (13.3)	−4.0 (12.3)	<0.001	234.0 (185.9)
HbA1c	261	5.9 (1.2)	212	6.0 (1.2)	5.6 (0.7)	0.4 (0.8)	<0.001	254.2 (200.1)
HbA1c by glycaemic control classification	261		212					
Normoglycemia HbA1c: <5.7%	158 (60.5)	5.2 ±0.3	110 (51.9)	5.3 (0.2)	5.2 (0.3)	−0.1 (0.2)	0.010	252.6 (212.0)
Pre-diabetes HbA1c: 5.7–6.4 %	55 (21.1)	6.0 (0.2)	54 (25.5)	6.0 (0.2)	5.7 (0.3)	−0.3 (0.3)	<0.001	242.4 (169.3)
Diabetes HbA1c: ≥6.5%	48 (18.4)	7.9 (1.3)	48 (22.6)	7.9 (1.3)	6.5 (0.9)	−1.4 (1.3)	<0.001	271.4 (206.9)
Fasting Glucose (mmol/L)	211	6.3 (3.5)	186	6.1 (1.9)	5.9 (1.3)	−0.3 (1.3)	0.009	246.0 (202.8)
Insulin (mU/L)	234	12.5 (8.5)	214	12.6 (8.5)	10.7 (7.1)	−1.9 (6.8)	<0.001	241.0 (191.0)
Triglycerides (mmol/L)	258	1.4 (0.8)	237	1.4 (0.9)	1.2 (0.5)	−0.2 (0.7)	<0.001	243.5 (192.3)
Total Cholesterol (mmol/L)	255	5.5 (1.5)	231	5.5 (1.3)	5.6 (1.8)	0.2 (1.1)	0.013	240.7 (188.7)
HDL-Cholesterol (mmol/L)	255	1.4 (0.4)	230	1.4 (0.4)	1.5 (0.4)	0.10 (0.22)	<0.001	241.2 (188.9)
LDL Cholesterol (mmol/L)	250	3.45 ± 1.36	226	3.42 ± 1.22	3.60 ± 1.63	0.18 ± 1.02	0.009	241.5 ± 190.1
Liver Function								
ALT (U/L)	267	28.8 (27.7)	250	29.1 (28.5)	22.7 (11.7)	−6.4 (27.1)	<0.001	253.9 (205.2)
AST (U/L)	262	24.0 (12.6)	245	24.0 (12.7)	20.5 (7.1)	−3.5 (12.6)	<0.001	256.8 (206.3)
GTT (U/L)	224	27.0 (23.0)	201	26.4 (19.0)	20.9 (12.4)	−5.6 (11.5)	<0.001	265.1 (216.3)

BMI—Body Mass Index, ALT—alanine aminotransferase, AST—aspartate aminotransferase, GTT—γ-glutamyltransferase, HbA1c—glycosylated haemoglobin, HDL—high density lipoprotein, LDL—low density lipoprotein.

**Table 4 nutrients-14-04406-t004:** Frequency of participants prescribed diabetes mediation and diabetes medications prescribed at comment and following program participation.

	Number of Participants, *n*		Number of Medications Prescribed, *n*	
	Prescribed Initial Consultation	Prescribed Post Program	Prescribed Initial Consultation	Prescribed Post Program
Sulfonylurea	11	4	11	4
SGLT-2	9	2	9	2
DDP-4	17	12	17	12
GLP-1	4	8	4	8
Metformin	56	47	57	47
Insulin	17	7	26	9
Diabetes medication—excluding metformin	42	26	67	35
All diabetes medication—including metformin	63	51	124	82

## Data Availability

Data described in the manuscript will not be made available because approval has not been granted by participants.

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
