# Peer review of "A Health Care Professional Delivered Low Carbohydrate Diet Program Reduces Body Weight, Haemoglobin A1c, Diabetes Medication Use and Cardiovascular Risk Markers—A Single-Arm Intervention Analysis"

_nutrients, 2022, doi:10.3390/nu14204406_

Round 1
Reviewer 1 Report
Brinkworth et al., have investigated the effects of consuming a low carbohydrate (LC) diet on weight loss and glycemic control. Their study shows that LC diet reduced body weight and HbA1C significantly but was accompanied by a small increase in plasma lipid levels.
This is an important question worth investigating given the global prevalence of obesity.
Here are a few specific comments:
Abstract:
Please do not use non-standard abbreviations in abstract such as BW.
Please expand LCD program
Introduction:
Please do not start a sentence with numbers.
Methods:
Is the Diversa Health Care Program a private for profit or non-profit government funded program? Please provide information. Also, see question related to conflicts of interest.
Was there a difference in magnitude of response between health coaches of various professions?
It is not clear whether the diet was provided by Diversa Health Program or only diet plan was provided and that participants had to buy or prepare their own diet. How did the authors ensure whether patients ate similar diet?
Can the authors provide samples of their typical diet plan?
What is the proportion of missing data in this study? Please include in your methods or results section.
Results:
While individual cardiovascular disease risk factors such as LDL, BP or glucose levels are important, can the authors please estimate CVD risk according to ACC/AHA guidelines
https://www.cvriskcalculator.com/
Risk based on initial visit and final visit.
This will add more value to your study.
Discussion:
Blood pressure: Line # 4, 2 mm Hg should be removed as it is related to DBP not SBP.
HbA1c: The discussion is lengthy. Please shorten.
Please tone down the statement ..:This suggest the possibility that the prescription of a LC diet…..magnified…..”
The authors may wish to discuss implications of increased lipid levels such as LDL in their discussion.
Conflicts of interest:
Did any authors receive compensation for their work from Diversa Health Program? If yes, please indicate.
Author Response
See letter attached

Reviewer 2 Report
Despite the growing threat of neoplastic diseases (especially respiratory) as an increasingly contributing source of morbidity and mortality in the world, cardiovascular (especially ischemic and cerebrovascular) pathology still leads as the primary cause of death in high-income countries. According to the WHO, in 2019, the top 10 causes of death accounted for 55% of the 55.4 million deaths worldwide. Moreover, still according to the WHO 2019 stats, diabetes has entered the top 10 causes of death, following a significant percentage increase of 70% since 2000. Diabetes is also responsible for the largest rise in male deaths among the top 10, with an 80% increase since 2000.
In terms of more than mere cost-efficiency, prevention still stands as a major weapon for promoting health and good quality of life for those subsets of individuals particularly exposed to the vast array of risk factors already identified for cardiovascular diseases and diabetes. For both of these entities, prevention starts from lifestyle and/or diet changes, according to specific biochemical or physical endpoints to be reached. The purpose of this study was to assess the effectiveness of the Diversa Health Care Program (a paid subscription-based health service that delivers a personalised LC plan by HCPs including general practitioners, health educa- tors and allied health professionals who provide education, medical monitoring, and supervised care via telehealth) for the management of body weight and metabolic and diabetes risk factors. Specifically, the primary endpoints to assess effectiveness of the program were changes in body weight and glycosylated haemoglobin (HbA1c) at the participants final recorded program consultation visit. Secondary endpoints assessed in- cluded blood pressure, waist circumference and lipid analyses.
The methods implemented for the study structure are well presented, describing the single-arm retrospective model through which the patient population has been analyzed. Inclusion and exclusion criteria are transparently depicted as well, with some emphasis given to the regularity and time span in between professional consultations, in an attempt to raise the scientifical proof of value of the diet program itself as a potential tool for global prevention. Also for the same reason, it was necessary for the authors to describe the structure of the Diversa Health Care Program. All statistical analyses were conducted using SPSS statistical software (Version 28.0, Armonk, NY), with a significance p threshold of <0.05 for extra accuracy.
Results are offered to the reader in multiple forms of presentation, from plain text to tables and diagrams, in order to better accommodate the need for detailed insights on any given statistics. The Diversa Health Care Program’s hypothesis, firstly stated in the introduction paragraph and then hereby sustained statistically, somehow imposes itself against the traditional dietary recommendations dispensed so far for subjects at risk, with an energy reduced, high carbohydrate, low fat (HC) diet for obesity and T2D management. Nevertheless, the analysis hereby presented is also sustained by a consistent body of evidence made up from systematic reviews and meta-analysis that also recorded improvements in therapeutic end-points such as waist circumference, blood glucose and medication necessary in order to achieve disease control.
The potential implications in terms of worldwide healthcare and global prevention are obviously massive, with better parameters recorded after intervention for body weight, diabetes balance, weight loss, metabolic risk factors as blood pressure, blood lipids and liver function markers; yet the fact that the study was financed by the Diversa Health Pty Ltd itself poses some limitations in terms of credibility, regardless of the lack of disclosure for any conflict of interest. The possibility that certaingroups of patients may benefit from a low carb diet programbetter than others discloses an additional area of desirable further insight for future researches, given the character of broad generalizability of the present one. Finally, longer follow-up intervals are needed to confirm sustainability and durable positive health effects on the target population.
Author Response
See letter attached

Reviewer 3 Report
The authors described that HCP delivered LC diet program is feasible and effective for achieving clinically relevant weight loss, improvements in blood glucose control and reductions in cardiovascular disease risk factors for individuals who are overweight or obese and/or with elevated blood glucose levels.
Please can describe the role of physical exercise, the gender and depressive symptoms on the diet program?
To this regard there are some examples that LPD ameliorate the renal function and inflammatory state but is related to depressive symptoms.
Author Response
See letter attached

Round 2
Reviewer 3 Report
it is ok for me